# Evaluating sickness absence duration by musculoskeletal and mental health issues: a retrospective cohort study of Scottish healthcare workers

Evangelia Demou,[1] Shanley Smith,[2] Abita Bhaskar,[1] Daniel F Mackay,[2] Judith Brown,[2] Kate Hunt,[1] Sergio Vargas-Prada,[3,4,5,6] Ewan B Macdonald[2]

[1]MRC/CSO Social and Public Health Sciences Unit, University of Glasgow, Glasgow, UK
[2]Public Health, Institute of Health and Wellbeing, University of Glasgow, Glasgow, UK
[3]Center for Research in Occupational Health (CiSAL), Universitat Pompeu Fabra, Barcelona, Spain
[4]Unidad Central de Contingencias Comunes (U3C), ASEPEYO, Barcelona, Spain
[5]CIBER Epidemiología y Salud Pública (CIBERESP), Madrid, Spain
[6]IMIM (Hospital del Mar Research Institute), Barcelona, Spain

**Correspondence to**
Dr Evangelia Demou;
evangelia.demou@glasgow.ac.uk

## ABSTRACT

**Objectives** Sickness absence (SA) among healthcare workers is associated with occupational and non-occupational risk factors and impacts employee health, healthcare delivery and patient health. At the same time, healthcare is one of the employment sectors with the highest rates of work-related ill health in the UK. Musculoskeletal (MSK) and mental health (MH) issues are leading causes of SA, but there is a lack of research on how certain MSK/MH conditions impact on SA duration. The study aim is to determine differences in SA duration by MH and MSK disorders in healthcare employees.

**Methods** Survival analyses were used to estimate SA duration due to MSK and MH problems over 6 years, and Cox's proportional hazards models to determine the HRs of returning to work, using a bespoke Scottish health board database with over 53 000 SA events. SA duration and time to return-to-work (RTW) were estimated for employees by age, gender, job and health conditions.

**Results** MSK and MH conditions accounted for 27% and 6% of all SA events and 23.7% and 19.5% of all days lost, respectively. Average SA duration was 43.5 days for MSK and 53.9 days for MH conditions. For MSK conditions, employees with low back or neck pain had the fastest RTW (median $P_{50}$: 7 days), whereas employees absent due to depression took the longest ($P_{50}$: 54 days). The most influential sociodemographic variables affecting RTW were age, gender and job category.

**Conclusions** Using a unique and rich database, we found significant differences in SA duration by presenting condition in healthcare workers. MH conditions, and depression specifically, accounted for the most working days' absence. Significant variations in duration were also observed for MSK conditions. Our findings can inform public health practitioners and healthcare managers of the most significant factors impacting MSK-related and MH-related SA to develop and implement tailored and targeted workplace interventions.

## INTRODUCTION

Sickness absence (SA) is a significant public health burden on government, employees, employers and public resources.[1–5] This is due to loss of productivity, increased

## Strengths and limitations of this study

► Healthcare is one of the employment sectors with significantly higher rates of sickness absence (SA), impacting on employee health, healthcare delivery and patient health. This study uses a unique, data-rich and bespoke database that allows for detailed assessment of the impact of musculoskeletal (MSK) and mental health (MH) conditions on absence duration and return-to-work (RTW) for healthcare employees of a Scottish health board.

► Cox's proportional hazards models applying time-varying coefficients were used to estimate SA duration due to MSK and MH problems over 6 years (n=48 007 unique SA events).

► The size and granularity of the database, giving six full years of SA and RTW data, and the range of variables collected–including demographic, job, SA start and end dates and self-reported conditions–allowed for the investigation of SA duration and time to RTW for healthcare employees by age, gender, job and health conditions.

► SA data are from a single large health board (approximately 12 000 employees) which limits the degree of generalisability. Also as SA absence is multicausal and it is necessary to consider an individual's work and workplace environment, there are a number of variables that are not collected as part of the service (eg, operational and organisational variables) which could potentially be important risk factors for SA and RTW in this population.

workload on other staff, as well as resources spent to cover incapacity to work and restore health.[2 6] SA data are increasingly being used as a measure of ill health.[7] The public health burden of SA is recognised, and governments and employers are developing policies towards providing support to employers and employees in improving health, reducing SA rates and improving return-to-work (RTW) times.[8–12]

Multiple factors need to be considered to effectively manage SA through policy and

practice.[13] SA rates vary by sector and employer, with the healthcare sector exhibiting one of the highest rates.[14] Additionally, within the health service, variations in SA rates depend on region, job category and salary grade, among other factors.[3 15 16] Cause of SA can have a major impact on duration and overall costs. Long-term SA (LTSA), although only responsible for a small percentage of SA *events*, makes up approximately 75% of absence *costs*, with musculoskeletal (MSK) and mental health (MH) disorders being two of the leading causes of LTSA.[2 3 14 17 18]

### Healthcare employees and SA

The healthcare sector is a diverse entity, which presents a unique set of challenges in relation to SA duration.[19] Studies show that healthcare workers experience high exposures to both occupational and non-occupational risk factors, negatively impacting on SA events.[19–24] Healthcare employee absence leads to increases in the risks to quality of patient care, workload stress of colleagues and employers' staffing costs.[19 25–28] The most recent 2016 figures from the Health and Safety Executive (HSE) report the healthcare industry as one of the employment sectors with the highest rates of work-related illness.[29] Ill health in the health and social care sector leads to around 4.8 million working days lost with the majority due to MH disorders, followed by work-related MSK disorders.[30] A study examining quality of care and treatment in 14 English hospital trusts found a positive association between the inpatient to staff ratio and high hospital standardised mortality ratio scores[31] and also identified insufficient nursing establishments and poor staffing levels on night shifts and weekends, partially due to high SA rates.[31]

### Impact of SA cause and duration

Certain health conditions result in longer periods of SA.[14] These thus have large potential interest as the focus of workplace interventions, due to the greater impact longer SA spells have on individuals, colleagues and employers.[32 33] There are potential long-term effects on individuals who experience particular types of SA events; one Swedish cohort study, for example, found an association between long-term absence and lower disposable income in the 2–6 years following the absence event.[34] A previous study investigating the effectiveness of an early SA management service in a Scottish Health Board found that although the leading causes of SA were gastrointestinal (GI) problems, followed by cold/cough/influenza (CCI), the greatest impact on total number of days lost was due to MSK and MH problems.[18]

### MSK-related absence

Health Response UK reports MSK injury as the leading cause of absenteeism and related cost to UK industry, with back pain as one of the most common causes of MSK-related SA.[17] Back pain is often recurrent, and the majority of people who have activity-limiting back pain go on to have recurrent SA episodes.[35] A systematic review examining SA and RTW found that approximately 20% of employees with back pain have some SA episodes in the 6-month to 12-month period following the back pain episode. RTW in this review was estimated at up to 1 month for 68% of individuals, 85% of individuals were back to work between 1 and 6 months and 93% were back to work at 6 months post the SA event.[36] Although RTW rates for back pain tend to be high, the recurrent nature of back pain increases the risk for substantial overall impact on work days lost through repeated periods of SA.

Neck problems are also a common MSK condition resulting in lost working days.[17] One cohort study investigating the relationship between physical and psychosocial features of the workplace and SA found that work activities involving neck flexion and neck rotation, high job demands, low skill discretion and low job security were significantly associated with SA due to neck pain.[37]

### MH-related absence

A number of systematic reviews highlight the importance of recognising MH issues in the workplace to assist in the reduction of associated SA.[38 39] However, there is limited evidence on how certain MH conditions affect RTW times.

Depression is a leading cause of MH-related absence.[14] The extent and severity of depressive symptoms, comorbidity of anxiety, social and emotional support, education and long symptom periods prior to diagnosis can affect the course of depression and RTW times.[40–42]

For those experiencing anxiety that results in SA, previous anxiety episodes, older age, lower education levels and long durations of untreated and undiagnosed symptoms contribute to longer absences.[42]

Detailed RTW figures following LTSA are scarce. Estimates suggest that as few as 13% of people who have experienced depression and anxiety are in employment, compared with 33% of people experiencing other chronic illnesses.[43]

### Sociodemographic and occupational factors and SA

Several studies have investigated the associations between SA and an employee's sociodemographic and socioeconomic characteristics.[19 44–48] Evidence shows that SA is multicausal, and that in addition to providing support for an individual's specific characteristics and incapacitating condition, it is necessary to also consider an individual's work and workplace environment in order to effectively manage and improve RTW times.[49 50]

### Aim

Despite a great deal of research recognising the prevalence of MSK and MH issues and their impact on SA rates[3 19–24], there remains a lack of information on how certain types of MSK and MH conditions may affect SA duration, which is vital to inform and improve current and new SA interventions, as well as support workplace modifications for SA prevention. This study aims to examine the impact of certain MSK and MH conditions on SA duration in healthcare workers, using a bespoke

database[18 51] over a 6-year period, and to assess how health, sociodemographic factors, work characteristics and occupational factors affect SA duration.

## METHODS
### Study population
The study population includes all participants in the EASY (Early Access to Support for You) SA management service of a Scottish health board.[18 51] EASY is a telephone-based service that provides early intervention (from day 1 of absence) based on the biopsychosocial model.[51 52] Data on all staff who voluntarily engage with EASY (compliance rate was approximately 80%) are routinely entered into a bespoke database, including age, gender, job family, self-reported cause of absence, date of first day of absence (FDA) and RTW date.[18 51] Detailed descriptions of the EASY service can be found elsewhere.[18 51] We examined data on 66 490 unique absence events recorded in the EASY database between May 2008 and December 2014.

### Defining and recoding variables
SA duration was calculated from the difference between the first date of absence and the RTW date. Cause of absence was grouped into seven categories: respiratory; MSK; MH; GI; ear, nose and throat; CCI and all others. Conditions assigned to the MSK category were grouped into eight subgroups according to anatomical site: hip, knee, low back, lower limb, neck, shoulder, upper limb and all others. MH conditions were allocated to eight subgroups according to diagnostic group: bipolar disorder, anxiety, depression, schizophrenia, panic attacks, self-harm, stress and all other psychiatric disorders.

After missing data were eliminated (<0.05%), four main exclusion criteria were applied: (1) if there was no FDA (n=196), (2) if the FDA was a Saturday or Sunday (n=4881), as there was no SA service on those days and absences would not be recorded on FDA, (3) if the 'date opened' (ie, the date the EASY service contacted the absentee) was before the FDA (n=828) and (4) if the 'date opened' was equal to or after the RTW date (n=3465). Due to overlaps among the missing data and exclusion criteria, a total of 13 286 absences were excluded, giving a total of 53 193 unique absence events; hence, analysis was based on these 53 193 unique absence events.

### Analytic strategy
#### Descriptive statistics
Descriptive statistics and crosstabs were produced for the entire EASY population, which included all causes of SA (n=53 193). Descriptive statistics and crosstabs were then produced for two specific causes, MH (n=3093) and MSK (n=6969) conditions.

Mean absence duration was calculated for the entire EASY population in the study, the entire EASY population without MSK and MH (n=43 131), MSK only, MH only, MSK subcategories and MH subcategories for each

year from May 2008 to December 2015. The data were divided into 7 years as follows: May 2008–April 2009 (Year 1), May 2009–April 2010 (Year 2), May 2010–April 2011 (Year 3), May 2011–April 2012 (Year 4), May 2012–April 2013 (Year 5), May 2013–April 2014 (Year 6) and May 2014–December 2015 (Year 7). Absences commencing in Year 7 were not included as the year was incomplete and in numerous cases, absence could still be ongoing; hence, further analyses presented were based on 48 007 unique absence events.

### Kaplan-Meier survival analysis and Cox proportional hazards model
Absence duration was analysed using Kaplan-Meier survival curves and Cox's proportional hazards models to determine the HRs of absentees returning to work. We tested the assumption of proportional hazards using Schoenfeld residuals and where violated, we corrected for this by adding the relevant time-varying covariate (online supplementary table S1). The model takes into account each SA event as well as individuals with multiple absence events by calculating cluster robust standard errors and the multivariate model controlled for several occupational and individual variables including gender, age, job family, job type, cause of absence, day of absence, season of absence and year of absence (online supplementary tables S2a–f).

All statistical analyses were conducted using STATA V.12.0 and R V.3.1.1.

## RESULTS
Our sample contains 11 694 individual people and 48 007 absences (May 2008–April 2014). Descriptive statistics for our sample (May 2008–December 2014; n=48 007) are reported in table 1. Almost 9 out of 10 (87.9%) absences were from female employees (n=42 187). Over the 6-year period investigated, the three most common causes of SA events are GI (n=13 459); CCI (n=8657) and MSK (n=6530) problems. Almost half the absences (45.3%) were from the nursing/midwifery job category, and a further 20% in administrative services. Numbers in part time (43.2%) and full-time employment (56.8%) were relatively similar.

Fewer of those with MSK-related absences (n=6530) were in the two youngest age groups, than in the total population, and the proportion of those with MSK absences who were in the nursing/midwifery (n=3428) job category (52.5%) was even higher than seen in the total population (45.3%). In the MH population, 91.8% are women, compared with 87.9% of the total population (table 1).

### Mean duration (in days) of absence by cause of sickness
GI and CCI problems account for the largest number of SA *events*, 28% and 18%, respectively. However, figure 1A shows that the *impact*, in number of days absent, these causes have on total number of working days lost is much

**Table 1** Descriptive statistics for absences in the total EASY population and MSK and MH subgroups

|  | Total sample (N=48 007) | | Total MSK absences (N=6530) | | Total MH absences (N=2921) | |
|---|---|---|---|---|---|---|
|  | n | % | n | % | n | % |
| **Age group** | | | | | | |
| 16–29 years | 6885 | 14.34 | 588 | 9.00 | 253 | 8.66 |
| 30–39 years | 10 772 | 22.44 | 1124 | 17.21 | 639 | 21.88 |
| 40–49 years | 15 257 | 31.78 | 2291 | 35.08 | 1102 | 37.73 |
| 50–59 years | 13 003 | 27.09 | 2181 | 33.40 | 813 | 27.83 |
| 60+ years | 2090 | 4.35 | 346 | 5.30 | 114 | 3.90 |
| **Gender** | | | | | | |
| Male | 5820 | 12.12 | 963 | 14.75 | 241 | 8.25 |
| Female | 42 187 | 87.88 | 5567 | 85.25 | 2680 | 91.75 |
| **Job family** | | | | | | |
| Administrative services | 9597 | 19.99 | 1003 | 15.36 | 550 | 18.83 |
| Allied health profession | 4914 | 10.24 | 502 | 7.69 | 208 | 7.12 |
| Healthcare sciences | 2130 | 4.44 | 284 | 4.35 | 105 | 3.59 |
| Manager | 150 | 0.31 | 13 | 0.20 | 5 | 0.17 |
| Medical and dental | 1301 | 2.71 | 125 | 1.91 | 33 | 1.13 |
| Medical and dental support | 876 | 1.82 | 83 | 1.27 | 35 | 1.20 |
| Nursing/midwifery | 21 734 | 45.27 | 3428 | 52.50 | 1614 | 55.26 |
| Other therapeutic | 2100 | 4.37 | 193 | 2.96 | 74 | 2.53 |
| Personal and social care | 443 | 0.92 | 56 | 0.86 | 18 | 0.62 |
| Support services | 4762 | 9.92 | 843 | 12.91 | 279 | 9.55 |
| **Job type** | | | | | | |
| Part time | 20 758 | 43.24 | 2871 | 43.97 | 1431 | 48.99 |
| Full time | 27 249 | 56.76 | 3659 | 56.03 | 1490 | 51.01 |

EASY, Early Access to Support for You; MH, mental health; MSK, musculoskeletal.

less (11.8% for GI and 6.3% for CCI). The health conditions with the highest impact on total number of working days lost are MSK (24%) and MH (20%) conditions. Mean absence duration ranged from 5.6 days for CCI-related to 53.3 days for MH-related absences. MSK absences had an overall mean duration of 28.9 days.

The three most common types of MSK problems in this population are low back pain (33.6% of all SA events), lower limb (9%) and upper limb problems (9%). Figure 1B shows that, within the subgroup with absences due to MSK, low back pain had the highest per cent impact on total number of working days lost (26.8%), followed by upper limb problems (12.3%) and lower limb problems (9.1%). Mean absence duration within the MSK-related absences ranged from 17 days due to neck problems to 40 days for upper limb. All other MSK conditions did not differ greatly in duration and ranged from 23 to 34 days.

The three most common types of MH problems among SA episodes in the EASY population are stress (64.8%), anxiety (15.9%) and depression (13.7%). The job categories with the highest cases of MH-related absences are nursing/midwifery (55.3%) and administrative services (18.8%). Figure 1C demonstrates that stress accounts for the largest per cent of working days lost (62.7%),

followed by depression (18.9%) and anxiety (14.6%). There was no significant change year on year in mean duration of MH-related absences (online supplementary figures S1a–c), with depression resulting in the longest absences in all years (overall mean duration of 72.1 days). Anxiety-related and stress-related absences had a mean duration of 48.1 and 50.7 days, respectively.

Mean absence duration in Years 2, 3, 4, 5 and 6 of the EASY service was compared with Year 1 using linear regression for all causes of SA to examine potential significant changes over time and then for each of the nine conditions (online supplementary figure S1a). Compared with Year one, mean absence duration for MSK cases in Year 2, Year 3 and Year 4 was significantly lower (online supplementary figure S1a); mean absence duration for GI cases in Years 5 and 6 was also significantly lower and mean absence duration for CCI cases in Years 2, 5 and 6 was significantly lower (online supplementary figure S1a).

**SA duration and RTW**
Figure 2 show the Kaplan-Meier RTW curves for all SA events minus MSK-related and MH-related absences (figure 2A), for the MSK-related absences (figure 2B and online supplementary figure S2) and for the MH-related

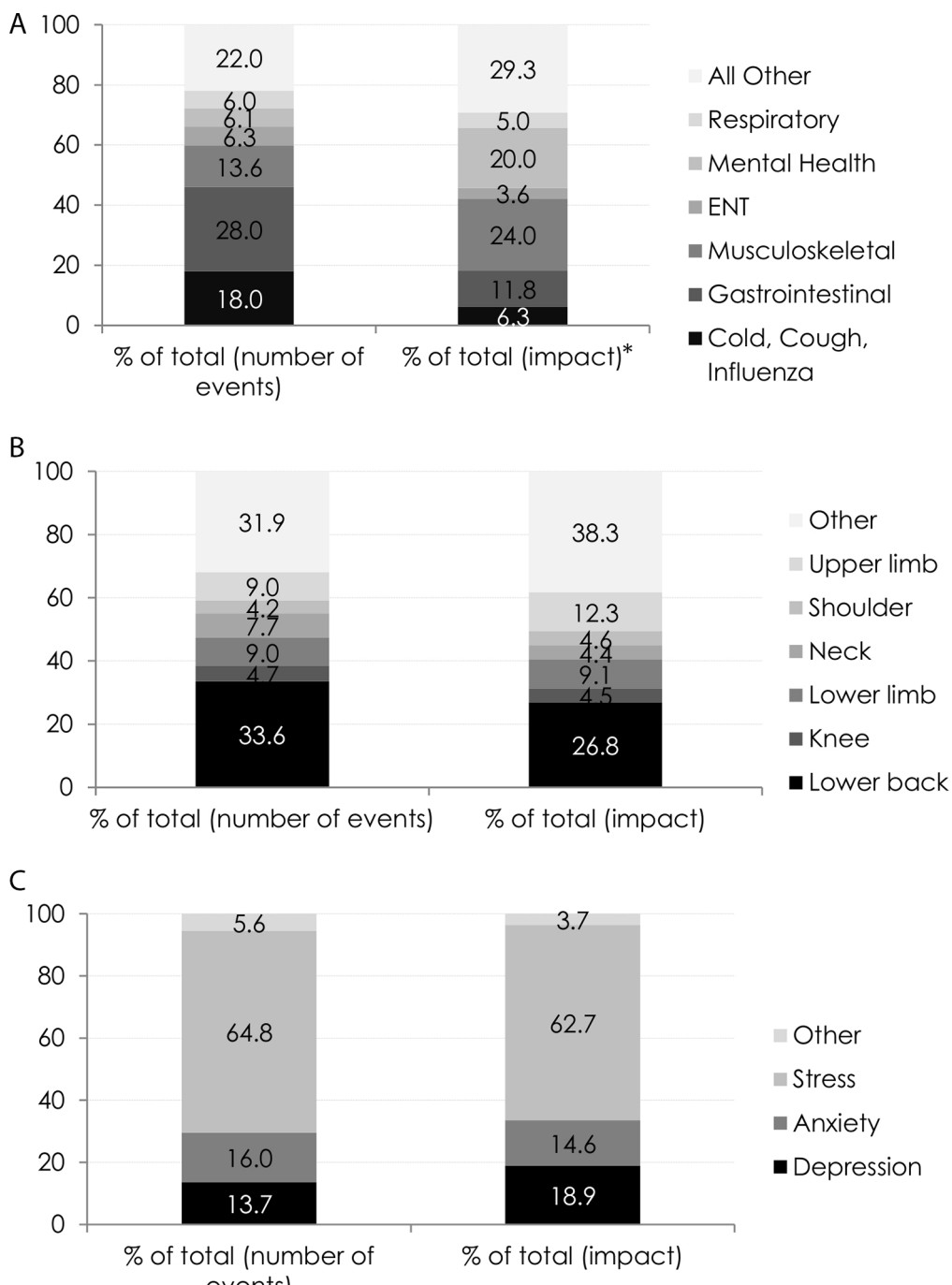

**Figure 1** Cause of sickness absence (percentage of total number of events and percentage of impact) for (A) all conditions, (B) MSK conditions** and (C) MH conditions***. *Impact is estimated as per cent impact and calculated by number of events times the average condition-specific absence duration divided by the sum of impact for all causes times 100. **Number of absences due to 'hip' were too small and grouped into 'other' category. ***Number of absences due to 'bipolar disorder', 'schizophrenia', 'panic attacks' and 'self harm' were too small and grouped into 'other' category. ENT, ear, nose and throat; MH, mental health; MSK, musculoskeletal.

absences (figure 2C and online supplementary figure S3). RTW for staff absent because of MH problems was much longer than all other causes of absences (figure 2A). For example, 50% for staff absent from work due to an MH problem had returned to work by 35 days (median), whereas 50% of those with an absence due to an MSK condition or all other conditions had returned within 10 and 5 days, respectively, of their FDA. As shown in

figure 2B, there are significant differences in RTW duration by subconditions within absences due to an MSK condition. Upper limb conditions result in the longest absences (50% of staff RTW by 25 days), whereas lower back and neck problems result in the shortest absences (50% of staff RTW by 7 days for both conditions). For the other MSK conditions (knee, lower limb, shoulder and other), 50% of the population RTW ($P_{50}$) between

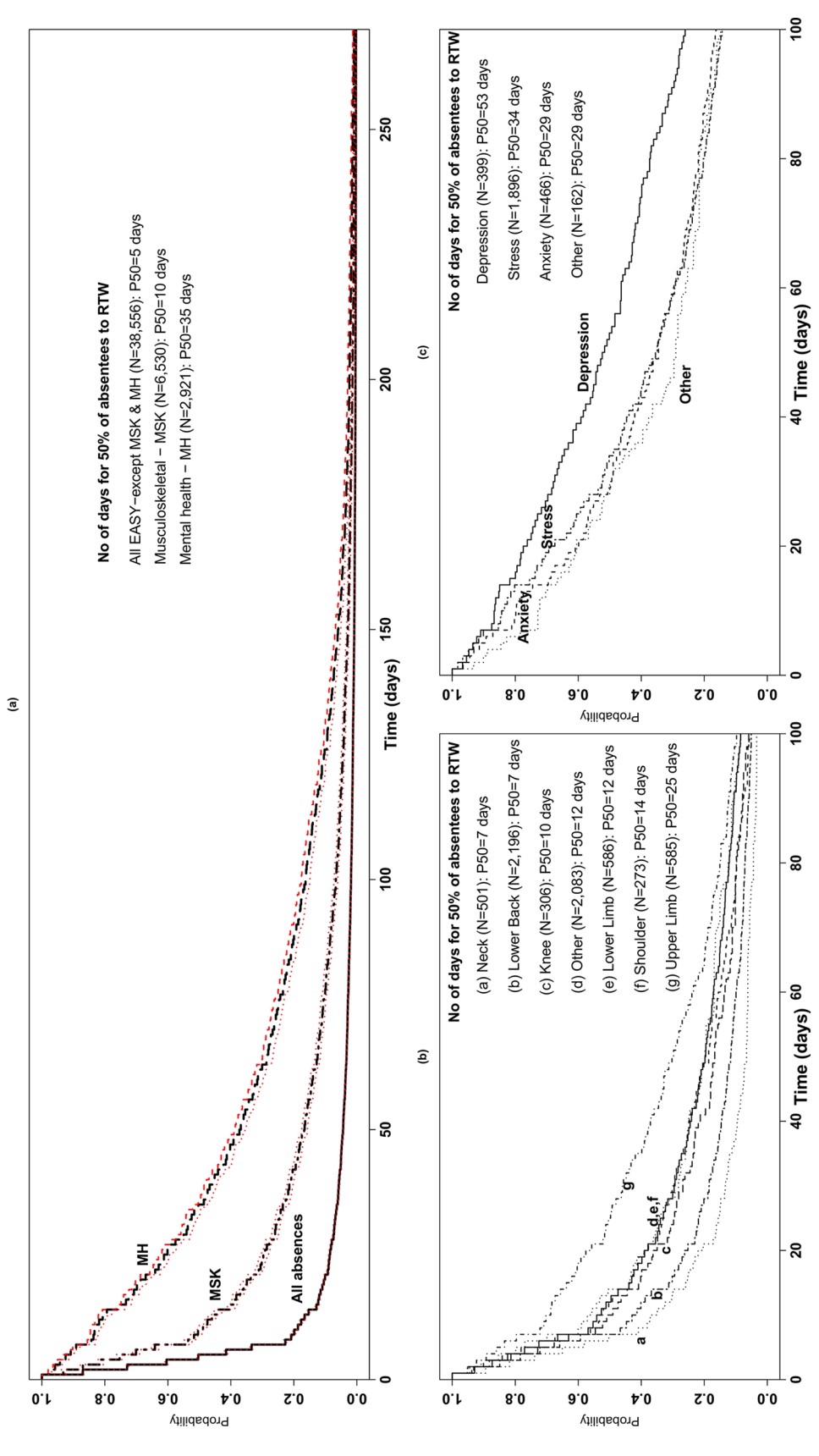

**Figure 2** (A) Return-to-work curves for (top) all absences with 95% CI; (bottom-left) (B) MSK-related absences by MSK condition and (bottom-right) (C) MH-related absences by MH condition (for survival curves with 95% CIs for MSK and MH conditions; see online supplementary figures S2 and S3. EASY, Early Access to Support for You; MH, mental health; MSK, musculoskeletal; RTW, return-to-work.

10 and 14 days. MH-related absences are much longer (figure 2C). Depression is the leading cause of longer SA events, with 50% of staff RTW by 53 days, followed by stress (50% of staff RTW by 34 days) and anxiety and other MH conditions (50% of staff RTW by 29 days).

## Multivariate analysis for all EASY absences (minus MSK and MH), MSK and MH

Table 2 shows the results of the multivariate Cox regression analysis to investigate the differences in the risk of RTW between the three main absence groups, gender, job title and year of absence. In the analysis for the entire population, the proportional hazards assumption was not met for the population, sex, age, job category, job type and day or year variables (see online supplementary table S1). For the MSK injuries, the population, sex and job category variables violated the proportional hazards assumption, whereas for the MH analysis, the population and day variables violated it, and, therefore, the analyses were rerun including these variables as time-varying coefficients (table 2 and online supplementary tables S2a–f).

In terms of absence cause, in comparison with all of EASY SA episodes (minus episodes due to MSK and MH), the risk of RTW was 52% (HR 0.48; 95% CI 47 to 0.50, P<0.0001) and 77% (HR 0.23; 95% CI 22 to 0.24, P<0.0001) longer for absences due to MSK and MH conditions, respectively. No real differences were seen after adjusting for gender, age, job family, job type, cause of absence, day of absence, season of absence and year of absence.

For MSK-related absences, staff absent due to neck problems were at risk of being off 17% longer (HR 1.17; 95% CI 1.05 to 1.30, P=0.006) compared with those absent because of lower back problems, whereas employees who were absent because of all other MSK conditions (HR 0.72; CI 0.67 to 0.78) had quicker RTW.

The risk of returning to work after being absent due to anxiety, stress and all other MH-related absences was significantly higher, when compared with depression, and was 64% (HR 1.64; 95% CI 1.39 to 1.93), 65% (HR 1.65; 95% CI 1.39 to 1.95) and 78% (HR 1.78; 95% CI 1.38 to 2.30) of longer duration, respectively (all P<0.001).

The analysis by gender demonstrated that for all absences (minus MSK and MH) and for MSK absences, women exhibited longer RTW times than men (table 2). No significant gender difference in RTW was observed for MH conditions. From the data, it is not possible to ascertain whether full-time or part time working has any association with RTW, as hours and days of employment were not included in the database.

In terms of job category, for all conditions (minus MSK and MH), nurses had the longest SA duration prior to RTW followed by staff in 'support services' (table 2). Employees in 'other therapeutic services' and 'medical and dental' staff demonstrated 52% (HR 1.52; 95% CI 1.43 to 1.63) and 45% (HR 1.45; 95% CI 1.34 to 1.58), respectively, shorter times to RTW than staff in 'nursing/midwifery'. For MSK-related absence, the only differences

to the reference category 'nursing/midwifery' were observed for 'support services' (HR 1.07; 95% CIs 0.99 to 1.16, P=0.103) and 'medical and dental' (HR 1.24; 95% CI 0.98 to 1.57, P=0.073), but these fell just short of conventional levels of significance. For MH-related absences, only 'support services' (HR 1.21; 95% CI 1.05 to 1.40, P=0.009) staff had significantly shorter time to RTW compared with 'nursing/midwifery' staff.

The analysis by year of absence using Year 1 as the reference category showed that staff took significantly longer to RTW in Years 3–6 for all conditions (minus MSK and MH). Among MSK absences, in Years 2–4, staff time to RTW was significantly shorter than for Year 1. There were no significant differences in time to RTW by year for staff absences due to an MH condition. Additionally, when examining time to RTW by day of absence start, absences starting on a Tuesday and Friday took significantly longer to RTW (online supplementary tables S2a–f). No significant differences in time to RTW were observed for season in which the absence occurred (P>0.05) (online supplementary tables S2a–f).

## DISCUSSION
### Summary of findings
We found significant differences in SA duration by presenting condition in a population of healthcare workers. MH conditions, and depression specifically, accounted for the most working days' absence. We also observed significant variations in duration for different MSK conditions. Upper limb disorders resulted in the longest SA durations among these healthcare workers, with 50% of staff returning to work by day 25. Lower back and neck problems resulted in the shortest MSK-related absences (50% RTW by 7 days), whereas knee, lower limb, shoulder and other MSK conditions resulted in SA durations that were comparable to each other (50% RTW between 10 and 14 days).

Employees within the nursing and midwifery job category accounted for almost half (45.3%) of all SA events recorded in the 6-year period and over half of the MSK-related (52.5%) and MH-related (55.3%) absence events. Nurses and midwives also had longer times to RTW than every other job category. In terms of gender, no significant differences were detected in RTW between men and women who were absent from work because of MH conditions, but for all other absence events, women took longer to RTW than men.

### Research in context to previous studies
The results of our study are in agreement with previous published work on the impact of MSK and MH conditions on SA and RTW.[2] [14] [53–55] A Danish cohort study aiming to identify prognostic factors associated with neck–shoulder pain resulting in LTSA found that pain intensity and job characteristics, such as heavy physical workload, were significantly associated with longer absence duration.[56] Armijo-Olivo et al (2016) also

**Table 2** Multivariate Cox regression RTW HRs for all SA episodes (minus MSK and MH), MSK and MH conditions with time-varying coefficients

| | All conditions (minus MSK and MH) | | MSK conditions | | MH conditions | |
|---|---|---|---|---|---|---|
| | HR (95% CI) | P | HR (95% CI) | P | HR (95% CI) | P |
| **Population** | | | | | | |
| All EASY except MSK and MH | 1 | | – | – | – | – |
| MSK | 0.48 (0.47 to 0.50) | 0.000 | – | – | – | – |
| MH | 0.23 (0.22 to 0.24) | 0.000 | – | – | – | – |
| **MSK condition** | | | | | | |
| Lower back | – | – | 1 | | – | – |
| Knee | – | – | 0.83 (0.74 to 0.94) | 0.003 | – | – |
| Lower limb | – | – | 0.79 (0.72 to 0.86) | 0.000 | – | – |
| Neck | – | – | 1.17 (1.05 to 1.30) | 0.006 | – | – |
| Shoulder | – | – | 0.77 (0.68 to 0.87) | 0.000 | – | – |
| Upper limb | – | – | 0.60 (0.55 to 0.65) | 0.000 | – | – |
| Other | – | – | 0.72 (0.67 to 0.78) | 0.000 | – | – |
| **MH condition** | | | | | | |
| Depression | – | – | – | – | 1 | |
| Anxiety | – | – | – | – | 1.64 (1.39 to 1.93) | 0.000 |
| Stress | – | – | – | – | 1.65 | 0.000 |
| Other | – | – | – | – | 1.78 (1.38 to 2.30) | 0.000 |
| **Gender** | | | | | | |
| Male | 1 | | 1 | | 1 | |
| Female | 0.89 (0.87 to 0.94) | 0.000 | 0.74 (0.68 to 0.81) | 0.000 | 1.07 (0.93 to 1.23) | 0.374 |
| **Age** | 0.99 (0.99 to 0.99) | 0.000 | 0.99 (0.99 to 0.99) | 0.000 | 0.99 (0.99 to 1.00) | 0.000 |
| **Job category** | | | | | | |
| Nursing/midwifery | 1 | | 1 | | 1 | |
| Administrative services | 1.21 (1.17 to 1.25) | 0.000 | 1.23 (1.12 to 1.35) | 0.000 | 0.98 (0.88 to 1.08) | 0.647 |
| Allied health profession | 1.33 (1.28 to 1.39) | 0.000 | 1.19 (1.07 to 1.33) | 0.001 | 1.15 (1.00 to 1.33) | 0.057 |
| Healthcare sciences | 1.22 (1.15 to 1.29) | 0.000 | 1.31 (1.13 to 1.53) | 0.000 | 1.10 (0.88 to 1.38) | 0.419 |
| Manager | 1.36 (1.11 to 1.66) | 0.003 | 1.80 (1.05 to 3.07) | 0.033 | 1.46 (0.79 to 2.69) | 0.232 |
| Medical and dental | 1.45 (1.34 to 1.58) | 0.000 | 1.24 (0.98 to 1.57) | 0.073 | 1.51 (0.94 to 2.43) | 0.085 |
| Medical and dental support | 1.16 (1.06 to 1.27) | 0.002 | 1.40 (1.06 to 1.84) | 0.016 | 0.91 (0.61 to 1.35) | 0.631 |
| Other therapeutic | 1.52 (1.43 to 1.63) | 0.000 | 1.44 (1.19 to 1.75) | 0.000 | 1.24 (0.91 to 1.68) | 0.173 |
| Personal and social care | 1.14 (1.01 to 1.28) | 0.033 | 1.69 (1.26 to 2.27) | 0.000 | 0.78 (0.49 to 1.22) | 0.276 |
| Support services | 1.01 (0.97 to 1.05) | 0.642 | 1.07 (0.99 to 1.16) | 0.103 | 1.21 (1.05 to 1.40) | 0.009 |
| **Job type** | | | | | | |
| Part time | 1 | | 1 | | 1 | |
| Full time | 1.13 (1.10 to 1.16) | 0.000 | 1.04 (0.99 to 1.11) | 0.135 | 1.09 (1.00 to 1.18) | 0.044 |
| **Year** | | | | | | |
| May 2008–April 2009 | 1 | | 1 | | 1 | |
| May 2009–April 2010 | 0.98 (0.95 to 1.02) | 0.389 | 1.14 (1.03 to 1.26) | 0.014 | 0.99 (0.85 to 1.15) | 0.884 |
| May 2010–April 2011 | 0.95 (0.91 to 0.99) | 0.008 | 1.14 (1.03 to 1.26) | 0.013 | 0.99 (0.84 to 1.16) | 0.857 |
| May 2011–April 2012 | 0.93 (0.90 to 0.97) | 0.001 | 1.11 (1.01 to 1.23) | 0.034 | 0.91 (0.79 to 1.06) | 0.249 |
| May 2012–April 2013 | 0.91 (0.88 to 0.95) | 0.000 | 0.97 (0.88 to 1.08) | 0.627 | 0.89 (0.76 to 1.04) | 0.145 |
| May 2013–April 2014 | 0.95 (0.92 to 0.99) | 0.018 | 1.07 (0.97 to 1.19) | 0.159 | 1.03 (0.88 to 1.21) | 0.687 |

Continued

**Table 2** Continued

| | All conditions (minus MSK and MH) | | MSK conditions | | MH conditions | |
|---|---|---|---|---|---|---|
| | HR (95% CI) | P | HR (95% CI) | P | HR (95% CI) | P |
| TVC | | | | | | |
| Population | 1.01 (1.01 to 1.01) | 0.000 | 1.00 (1.00 to 1.00) | 0.000 | 1.00 (1.00 to 1.0) | 0.03 |
| Age | 1.00 (1.00 to 1.00) | 0.787 | | | | |
| Sex | 1.00 (1.00 to 1.00) | 0.012 | 1.00 (1.00 to 1.01) | 0.000 | | |
| Job category | 1.00 (1.00 to 1.00) | 0.000 | 1.00 (1.00 to 1.00) | 0.019 | | |
| Job type | 1.00 (1.00 to 1.00) | 0.003 | | | | |
| Year | 1.00 (1.00 to 1.00) | 0.268 | | | | |

EASY, Early Access to Support for You; MH, mental health; MSK, musculoskeletal; RTW, return-to-work; SA, sickness absence; TVC, time-varying coefficients.

demonstrated that occupation and health condition were significant factors, among others, in the rehabilitation process of people affected by MSK conditions.[57] Similarly for MH conditions, previous research suggests that SA associated with psychological ill health tends to be higher among National Health Service (NHS) healthcare workers than for other employment sectors in the UK.[58] This may be due to the pressured nature of the work, constant organisational changes and the large workload,[59] supporting the notion that organisational factors may contribute to the level of psychological ill health experienced by staff. These findings are also reflected in our study, where significant variations in absence duration are observed not only by certain MSK and MH conditions but also by job categories—with nursing and midwifery staff experiencing the majority of and longest SA events. One study looking at job family and SA in the healthcare sector reported that doctors had nine times lower rates of short-term SA and four times lower rates of LTSA, whereas nurses had three times lower rates of short-term SA when compared with other healthcare sector job families.[44]

### Strengths of the study

The latest HSE report lists healthcare staff as one of the categories with the highest SA rates and MH and MSK as leading causes.[14] This study is therefore particularly important as we are able to investigate in detail the durations of SA events by particular MSK and MH conditions in this population. The unique and rich EASY database[51] enables analysis of routine data collected in a systemic way across all job categories within the healthcare sector. While several studies have reviewed SA in healthcare settings, these have been limited in several ways. For instance, they examine a narrow range of healthcare workers, mainly doctors and nurses, or examine only broad categories of absence.[39] Thus, the size of the EASY database, giving six full years of SA and RTW data, and the range of variables collected–including demographic, job, start and end dates and self-reported conditions–are considerable strengths.

### Study limitations

While the availability of such rich data on routine SA is a major strength of this study, there are also some limitations to the data. This SA management service is only available from Monday to Friday, and, therefore, not all absences may have been recorded. To avoid any confounding, we removed absences starting on a weekend. The cause of absence is self-reported by the employee when they call the service and not based on a clinical diagnosis, and comorbidities–which may also impact on absence duration–are not collected.[53]

SA is multicausal, and it is necessary to consider an individual's work and workplace environment. However, the lack of granularity in SA data recorded is often the barrier into investigating the risk factors and causes impacting on SA duration. Linking bespoke databases, such as the one in this study, to personnel data or having uniformity in the variables collected across health boards would allow investigation of the impact of operational (eg, shift work) and organisational (region and structures) risk factors on SA rates and duration.

As this health board has a unique SA management service, the results may not be representative of all healthcare workers. However, a national standard requires all health boards in Scotland to work towards a 4% or less SA rate.[60] While this health board had a higher SA rate in early 2008, by the end of 2008, the SA rate had fallen to similar levels to the other health boards.[18] The latest data show that this health board's SA rate of 5.20% is in line with the Scottish NHS average of 5.16%.[60]

### Implications for policy and practice

Due to the financial and morale repercussions SA among healthcare employees has not only on healthcare staff themselves but also on their patients and employers,[31] it is an important focus of attention for healthcare management and for public health policies and practice reform. Recent systematic reviews have evaluated the effectiveness of different types of SA interventions and have found that multidisciplinary interventions involving collaboration

between employees, health practitioners and employers working to implement tailored modifications for the absentee were consistently more effective than generic non-tailored interventions targeted at all employees.[13 61]

## CONCLUSIONS

The results of this study further establish the need for occupational health, organisational and management interventions to address recognised individual and workplace stressors that can impact on SA duration. Our results suggest that employees with upper limb problems and depressive symptoms in particular could benefit from more tailored interventions to assist them in the RTW process following an SA event. A great burden of work loss due to both MSK and MH conditions was observed for nurses and midwives.

This research is important in terms of improving the health and well-being of NHS staff but may also improve the quality of patient care and subsequently public health. SA has far-ranging economic consequences for both employers and employees, as it simultaneously impacts on NHS resources/service delivery and on people's earnings if the SA is prolonged. These findings give a deeper insight into the link between health, organisational, operational and sociodemographic factors influencing SA. Understanding these relationships allows health providers to be better placed to plan the allocation of resources, build better models of SA management and inform the development of tailored SA interventions for NHS staff.

**Acknowledgements** The authors acknowledge Mark Kennedy, Keith Murray, Moyra Anderson and Raymond Hamill for their support.

**Contributors** ED developed the study and was the main author of the manuscript. AB and SS conducted the statistical analysis and contributed to the writing of the manuscript. DFM advised on the statistical analysis. DFM, JB, KH, SV-P and EBM reviewed and contributed to the manuscript.

**Funding** This work was supported by the Medical Research Council partnership grant MC/PC/13027 to ED and funding from the UK MRC and Chief Scientist Office (MC_UU_12017/12; SPHSU12) to AB and KH.

**Competing interests** None declared.

**Patient consent** Not required.

**Ethics approval** NHS Lanarkshire Research and Development (R&D) Management Approval (ref no L11071).

**Provenance and peer review** Not commissioned; externally peer reviewed.

**Data sharing statement** No additional data are available.

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
