## [Reviewer comments · BMJ Open]

ARTICLE DETAILS

TITLE (PROVISIONAL)	Evaluating Sickness Absence Duration by Musculoskeletal and Mental Health Issues. A retrospective cohort study of Scottish Healthcare Workers.
AUTHORS	Demou, Evangelia; Smith, Shanley; Bhaskar, Abita; Mackay, Daniel; Brown, Judith; Hunt, Kate; Vargas, Sergio; Macdonald, Ewan

VERSION 1 – REVIEW

REVIEWER	Daniel Olsson Unit of Medical Statistics, Department of Learning, Informatics, Management and Ethics (LIME), Karolinska Institutet, Sweden
REVIEW RETURNED	12-Jun-2017

GENERAL COMMENTS	2. Regarding the abstract: The methods section says that Survival analyses (a group of methods not a specific method) and Cox regression has been used, while in the results section only presents descriptive statistics. 3. While the study design may be appropriate, it is described as cross-sectional, while this is clearly a study following a cohort over time from first day of absence (FDA) to the return to work (RTW). 7. "The study aim is to determine differences in SA duration by MH and MSK disorders in healthcare employees". The descriptive statistics are in agreement with this aim, the Cox model is not. The authors state that the Cox model is used for estimating "hazards" and "SA duration". The Cox model is not suitable to estimate either of these quantities, but rather estimates ratios of hazards, it does not provide an estimate of the hazards themselves. I would have liked at least confidence intervals for the estimated durations. To adjust for potential confounders the authors could use quantile regression to estimate conditional quantiles of SA duration or use standardization to estimate marginal estimates. 11. The conclusion that the results can be used to develop and implement tailored and targeted workplace interventions for employees with MSK and MH problems, just by knowing the SA durations for different diagnoses, is not justified.
---

REVIEWER	Dr Serena Bartys University of Huddersfield, UK
REVIEW RETURNED	19-Jul-2017

GENERAL COMMENTS	This paper is focused on further understanding the duration of sickness absence due to mental health and musculoskeletal disorders in healthcare workers, with implications for appropriate interventions to tackle the burden and impact this may cause. This is an area of significant concern and worthy of further study - a particular strength of this paper is the availability of longitudinal data from a bespoke database (designed for the EASY study). There were some valuable findings in the study about the differences in SA between MH & MSK. However, I felt that these aspects were not clearly articulated in this paper upon first reading. The authors suggest there is a 'lack of research on the impact of specific MSK/MH conditions on SA duration' without clearly defining what 'impact' is - does this just mean duration? How is duration defined? Also, the use of the word 'specific' is misleading here, because when applied to certain health conditions, like MSK, e.g. specific and non-specific MSK, it relates to certain categories of MSK. Were the authors trying to highlight that sickness absence duration hasn't been examined (i.e. longitudinally) for these conditions separately? Some of these aspects were explained further in the manuscript, but the abstract needs amending to be more clearly focused - I would suggest that longitudinal research is lacking, and therefore to highlight this strength. Additionally, it wasn't clear from the abstract why healthcare workers had been selected as the target population - was this purely because the data had already been collected as part of the EASY study? It is suggested that SA is comparable to other similar organisations, but in what respect? Incidence, prevalence, duration? It needs to be made clearer that a SA intervention was implemented at the study site, which could have confounded the findings. In light of this, how representative are these findings? These issues need to have been discussed further and highlighted as a limitation. It wasn't clear until much further reading how many people were involved in the study, and whether the results were due to a small number of 'outliers', i.e. a few people taking the longest absence, as is the epidemiology of sickness absence. Was it possible to conduct a longitudinal cohort study to strengthen this aspect? Terminology needs to be consistent throughout, e.g. RTW times, or duration. I would suggest that the introduction could benefit from an updated literature review on this topic in general - some of the references used are now quite dated. More information is required on what the risk factors for SA are in the healthcare sector particularly, and how these are different to other sectors. Are these risk factors largely attributable to nursing staff, or is all kinds of jobs within the sector. Indeed, the data collected in this study come mainly from nursing staff - therefore, is it correct to talk about the healthcare sector, or should it just be type of job? There isn't enough information on organisational factors/influences presented that are typical to the healthcare sector other than nursing to make such a general claim.
---

	The authors suggest that sickness absence is multi-causal and it is necessary to consider an individual's work and workplace environment, but this isn't elaborated on in the results/discussion in terms of how these findings suggest such things can be addressed. Were there any analyses undertaken on those 20% of people who weren't included in the EASY study - could the inclusion of these people changed the results in any way? Which data were missing? Although these may have been small numbers, was there any pattern of missing data? The data selected for this study is a relatively small proportion of data collected in the EASY study - therefore a stronger rationale is needed as to why MH and MSK is selected. The numbers of people need to be documented earlier in the results section - where N is presented, it is referring to sickness absence events, and not the number of people. This links into the discussion about whether the 'impact' of SA is caused by a relatively small number of people. Indeed, the data indicates that most people had 1, relatively short absence. What does this mean for the effectiveness and cost-effectiveness of early interventions? Are they worthwhile and how can this be shown with these data? The discussion is under-developed and needs to bring in more relevant literature, and the nuances mentioned above. The study limitations section in particular needs expanding to take into account these queries/critiques, and also to evaluate choice of methods, etc. The conclusions and recommendations are vague and non-specific - there needs to be more information on how interventions would look based on these findings, and more recommendations for relevant policy.
--	--

VERSION 1 – AUTHOR RESPONSE

Reviewer: 1

Reviewer Name: Daniel Olsson

Institution and Country: Unit of Medical Statistics, Department of Learning, Informatics, Management and Ethics (LIME), Karolinska Institutet, Sweden

Please state any competing interests: None

Please leave your comments for the authors below

1. Regarding the abstract: The methods section says that Survival analyses (a group of methods not a specific method) and Cox regression has been used, while in the results section only presents descriptive statistics.

Author Response: All our survival analyses and Cox regression model results are presented in our results section (pages 13-17) and specific results are presented in our abstract as well (e.g. time-to-event for 50% of population, and most important factors for return to work).

2. While the study design may be appropriate, it is described as cross-sectional, while this is clearly a study following a cohort over time from first day of absence (FDA) to the return to work (RTW).

Author Response: We thank Reviewer 1 for pointing out this error. We accept the reviewer's comment and have corrected this.

3. "The study aim is to determine differences in SA duration by MH and MSK disorders in healthcare employees". The descriptive statistics are in agreement with this aim, the Cox model is not. The authors state that the Cox model is used for estimating "hazards" and "SA duration". The Cox model is not suitable to estimate either of these quantities, but rather estimates ratios of hazards, it does not provide an estimate of the hazards themselves. I would have liked at least confidence intervals for the estimated durations. To adjust for potential confounders the authors could use quantile regression to estimate conditional quantiles of SA duration or use standardization to estimate marginal estimates.

Author Response: We thank the reviewer for pointing out an error in the wording. The Cox model does not estimate the hazard or the duration, but as shown here in this paper the Cox models were used to estimate the hazard ratios of absentees returning to work. We have rechecked the entire document to ensure that no other typo in the wording is used throughout. Additionally, we have changed the wording in the abstract to clarify this point.

As suggested by the reviewer we have included confidence intervals in our survival analyses and amended the graphs. Figure 2a includes the CI lines, however as it would be too difficult for the reader we have included in the supplementary material figures of the individual survival analysis curves for each musculoskeletal and mental health condition separately. Please see amended Figure 2 and Figures S2 and S3 in the Supplementary material. Also as described in our methods sections as well as in the results (non-adjusted versus adjusted model results) we have controlled for several occupational and individual variables including: gender, age, job family, job type, cause of absence, day of absence, season of absence, and year of absence in our multivariate models.

We appreciate the Reviewer's suggestion of using other methodologies such as quantile regression to estimate conditional quantiles of SA duration. However, as stated the purpose of this study was to estimate duration of sickness and the hazard ratios of RTW for the healthcare workers involved (censored quantile regression models do not give hazard ratios). We have time to event for all RTW events and excluded all absences commencing in Year 7 as the year was incomplete and in numerous cases absence could still be ongoing. Quantile regression could be used if for instance the Cox assumption is not met, however, we had added a time-varying coefficient to correct for non-compliance to Cox assumption in our analysis. This has been clarified in the document in the methods section (see page 9, 4th paragraph). While we appreciate quantile regression is an alternative method to estimate conditional quantiles of SA duration, we have opted with the main methodology used in such studies (e.g. very recent studies on SA all use Kaplan-Meier survival analysis and Cox regression).

- Negrini et al. 2017. "How Can Supervisors Contribute to the Return to Work of Employees Who have Experienced Depression?" *Journal of Occupational Rehabilitation* (First Online: 29 June 2017)
- Olofsson T, et al. 2017. "Predictors of work disability after start of anti-TNF therapy in a national cohort of Swedish patients with rheumatoid arthritis: does early anti-TNF therapy bring patients back to work?" *Annals of the Rheumatic Diseases* 2017;76:1245-1252.
- Kausto J, et al. 2017. "Length of sickness absence and sustained return-to-work in mental disorders and musculoskeletal diseases: a cohort study of public sector employees." *Scand J Work Environ Health* 2017;43(4):358-366.
- Endo M, Sairenchi T, Kojimahara N, et al Sickness absence and return to work among Japanese stroke survivors: a 365-day cohort study. *BMJ Open* 2016;6:e009682. doi: 10.1136/bmjopen-2015-009682

4. The conclusion that the results can be used to develop and implement tailored and targeted workplace interventions for employees with MSK and MH problems, just by knowing the SA durations for different diagnoses, is not justified.

Author Response: We agree with the reviewer that just by knowing SA duration for different diagnosis is not the panacea for developing the most appropriate intervention. However, as we mention this is a first step in understanding the most important causes impacting on SA rates and SA durations within healthcare organisations, knowing and understanding the operational and organisational as well as the sociodemographic risk factors does inform the decision making process to identify the 'hot spots' which may need prioritising or tailored interventions (e.g. for nurses prioritise interventions for musculoskeletal disorders, or upper limb disorders as they have the longest time to RTW). We have further elaborated on this point to make this clear in the conclusion section (please see page 21; last paragraph)

Reviewer: 2

Reviewer Name: Dr Serena Bartys

Institution and Country: University of Huddersfield, UK Please state any competing interests: None declared

Please leave your comments for the authors below

This paper is focused on further understanding the duration of sickness absence due to mental health and musculoskeletal disorders in healthcare workers, with implications for appropriate interventions to tackle the burden and impact this may cause. This is an area of significant concern and worthy of further study - a particular strength of this paper is the availability of longitudinal data from a bespoke database (designed for the EASY study). There were some valuable findings in the study about the differences in SA between MH & MSK.

1. However, I felt that these aspects were not clearly articulated in this paper upon first reading. The authors suggest there is a 'lack of research on the impact of specific MSK/MH conditions on SA duration' without clearly defining what 'impact' is - does this just mean duration? How is duration defined? Also, the use of the word 'specific' is misleading here, because when applied to certain health conditions, like MSK, e.g. specific and non-specific MSK, it relates to certain categories of MSK. Were the authors trying to highlight that sickness absence duration hasn't been examined (i.e. longitudinally) for these conditions separately? Some of these aspects were explained further in the manuscript, but the abstract needs amending to be more clearly focused - I would suggest that longitudinal research is lacking, and therefore to highlight this strength. Additionally, it wasn't clear from the abstract why healthcare workers had been selected as the target population - was this purely because the data had already been collected as part of the EASY study?

Author Response: We have clarified as suggested by the Reviewer that we are describing the impact of musculoskeletal and mental health conditions on the duration of sickness absence in healthcare workers. We appreciate the reviewer's concern over the word 'specific' and that it could potentially be misinterpreted when discussing specific and non-specific MSK pain and have amended this to 'certain' throughout the manuscript. We have addressed the issue of highlighting the longitudinal dataset in the abstract (please see response to Reviewer 1, comment #3 above).

We have described why it is important to address sickness absence in health care workers due to potential impacts ill-health in this occupational group can have not only on employee health, but also on healthcare delivery and patient health. We have further strengthened this addressing the importance of this issue due to the fact that it is one of the industries with the highest rates of work-related ill-health in the UK based on the latest figures from the HSE (please see revised abstract and revised introduction page 5, 1st paragraph).

2. It is suggested that SA is comparable to other similar organisations, but in what respect? Incidence, prevalence, duration? It needs to be made clearer that a SA intervention was implemented at the study site, which could have confounded the findings. In light of this, how representative are these findings? These issues need to have been discussed further and highlighted as a limitation.

Author Response: What we have highlighted in our manuscript is that SA rates in this Scottish health board are comparable to the Scottish NHS average SA rates. Under our study limitations we have addressed the issue that this health board had a SA management intervention in place that allowed for the collection of relevant data. However, as already mentioned, over the same period, all Scottish health boards were required to reach the HEAT target of 4% SA rate as well as manage their SA rates. Therefore, other interventions and policies would have been applied to all other health boards as well (page 20 of original submission).

3. It wasn't clear until much further reading how many people were involved in the study, and whether the results were due to a small number of 'outliers', i.e. a few people taking the longest absence, as is the epidemiology of sickness absence. Was it possible to conduct a longitudinal cohort study to strengthen this aspect?

Author Response: We have included the number of individuals in our population (11,694) as well as unique sickness absence events over the 6 years in the first sentence of our results section.

4. Terminology needs to be consistent throughout, e.g. RTW times, or duration.

Author Response: we have checked and revised our manuscript to ensure consistency in the reporting.

5. I would suggest that the introduction could benefit from an updated literature review on this topic in general - some of the references used are now quite dated. More information is required on what the risk factors for SA are in the healthcare sector particularly, and how these are different to other sectors. Are these risk factors largely attributable to nursing staff, or is all kinds of jobs within the sector. Indeed, the data collected in this study come mainly from nursing staff - therefore, is it correct to talk about the healthcare sector, or should it just be type of job? There isn't enough information on organisational factors/influences presented that are typical to the healthcare sector other than nursing to make such a general claim.

Author Response: We agree with the Reviewer that some of our references in the introduction may be a bit old. These references are all highly relevant and we find appropriate to keep. As suggested by the Reviewer we have included some more recent references. These are:

- Hussey L, Turner S, Thorley K, et al. Work-related sickness absence as reported by UK general practitioners. *Occupational Medicine* 2012;62(2):105-11.
- Black C. Health at Work - an independent review of sickness absence. In: The Stationary Office, ed., 2011.
- Lammerts L, Schaafsma FG, Eikelenboom M, et al. Longitudinal Associations Between Biopsychosocial Factors and Sustainable Return to Work of Sick-Listed Workers with a Depressive or Anxiety Disorder. *Journal of occupational rehabilitation* 2016;26(1):70-9.
- van Vilsteren M, van Oostrom SH, de Vet HCW, et al. Workplace interventions to prevent work disability in workers on sick leave. *Cochrane Database of Systematic Reviews* 2015; Issue 10. Art. No.: CD006955.
- Burton A, Kendall N, McCluskey S, et al. Telephonic support to facilitate return to work: what works, how, and when? In: Department for Work and Pensions, ed. London, 2013.

The Reviewer is correct that a significant proportion of the SA events in this study come from nursing and midwifery staff (known in the literature to be the case). However, this is less than 50% of our total population of healthcare workers included in the study. The references we have included examine all healthcare workers (e.g. Boorman report, Franche et al. 2010, Gorman et al. 2010) as well as specific occupational groups (e.g. doctors- Murphy 2014). We have included SA risk factors reported for the healthcare sector such as region, job category and salary grade, among other factors and referenced highly cited reports addressing sickness absence throughout the NHS and not only for nurses. These include:

Wise J. Audit finds large variations in NHS staff sick days. *BMJ* 2011;342 doi: 10.1136/bmj.d953
Boorman S. NHS Health and Well-being - Final Report. Leeds: Department of Health; 2009.

6. The authors suggest that sickness absence is multi-causal and it is necessary to consider an individual's work and workplace environment, but this isn't elaborated on in the results/discussion in terms of how these findings suggest such things can be addressed.

Author Response: We have amended our limitations sections to discuss how in the future we would be able to address operational and organisational risk factors to sickness absence in healthcare workers. The barrier at this point in time is the data and non-uniformity of data that is collected across the healthcare sector. Please see the amended limitations section (page 20, 3rd paragraph).

7. Were there any analyses undertaken on those 20% of people who weren't included in the EASY study - could the inclusion of these people changed the results in any way? Which data were missing? Although these may have been small numbers, was there any pattern of missing data?

Author Response: As stated in our limitations section, this detailed data is collected as part of a sickness absence management service that operates Monday – Friday. Therefore, absences starting on a weekend may have not been recorded and these were excluded from all analysis.

Only a very small amount of data were missing (0.05%) and this was due to missing data on age, job family, FDA, RTW date. There was no pattern observed in the missing data.

We mention in the paper that the compliance rate for those engaging with the EASY service, and therefore had their data entered into the database was circa 80%. For the employees that did not engage with this service, no data was collected and therefore no analysis can be done (or inferences made). To clarify we have added in the description that the EASY service is a voluntary service and was not mandatory for employees to engage (please see page 8, 1st paragraph).

8. The data selected for this study is a relatively small proportion of data collected in the EASY study - therefore a stronger rationale is needed as to why MH and MSK is selected.

Author Response: We have described why it is important to address sickness absence in health care workers due to the potential impacts ill-health in this occupational group can have not only on employee health, but also on healthcare delivery and patient health. We have further strengthened the importance of this addressing this issue due to the fact that it is one of the industries with the highest rates of work-related ill-health in the UK based on the latest figures from the HSE (please see revised abstract and revised introduction page 5, 1st paragraph).

Furthermore, we have included another statement reiterating why MSK and MH related SA events are an important issue and warrant further research as ill health in the health and social care sector leads to around 4.8 million working days lost with the majority due to mental health disorders (2.1 million days lost), followed by work-related musculoskeletal disorders (1.2 million days lost). While the number of events is relatively small- compared to all SA events – the impact in terms of working days lost and knock on effects on employee wellbeing, service delivery and patient care are enormous.

9. The numbers of people need to be documented earlier in the results section - where N is presented, it is referring to sickness absence events, and not the number of people. This links into the discussion about whether the 'impact' of SA is caused by a relatively small number of people.

Author Response: We have added the number of individuals (please also see response to comment 3 above). Also we have ensured the groups are not too small –please see Figure 2 group sample sizes are noted.

10. Indeed, the data indicates that most people had 1, relatively short absence. What does this mean for the effectiveness and cost-effectiveness of early interventions? Are they worthwhile and how can this be shown with these data?

Author Response: The novelty of this study is that it investigates and quantifies the duration of sickness absence in healthcare workers by all absences as well as for certain types of musculoskeletal and mental health absence. To our knowledge this has not been done before, as the lack of such granularity of data is a barrier. The study and the data used alone cannot make any inference regarding the intervention used in this health board or other interventions and it is not possible to assess effectiveness or cost-effectiveness of the intervention.

The reviewer suggests that most absences are relatively short. However, 50% of all employees return to work in 5 days for all SA causes-except MSK and MH conditions, for MSK conditions this is 10 days and for MH 35 days. Extrapolated across the entire NHS (~1.5mil employees) this amounts to a significant number of working days lost, as well as loss in resources to bring in other staff to maintain appropriate levels of care. Therefore, understanding the epidemiology of SA in healthcare workers to inform the development of tailored interventions to try and reduce the number of days lost would have significant benefits not only for the health and wellbeing of the individual employee but for the NHS and the health of the public as well. We have expanded our conclusions section to further explain the significance and potential impact these findings could have in the development of tailored interventions that consider the individual, operational, organisational and condition specific factors that impact on SA duration (please see amended conclusions section).

11. The discussion is under-developed and needs to bring in more relevant literature, and the nuances mentioned above. The study limitations section in particular needs expanding to take into account these queries/critiques, and also to evaluate choice of methods, etc. The conclusions and recommendations are vague and non-specific - there needs to be more information on how interventions would look based on these findings, and more recommendations for relevant policy.

Author Response: As per both Reviewers comments we have amended both the limitation and conclusion section and have included more recent literature. Please see response to the above comments.

VERSION 2 – REVIEW

REVIEWER	Daniel Olsson Unit of Medical Statistics, Department of Learning, Informatics, Management and Ethics (LIME), Karolinska Institutet, Sweden
REVIEW RETURNED	10-Oct-2017

GENERAL COMMENTS	The authors show that different health conditions have different length of sickness absence and that these differences are relatively large in magnitude. This is by no means controversial results and only requires simple descriptive statistics to show. However, the authors' conclusions go far beyond what can be substantiated by the results of the study. The authors do not appear to understand the statistics presented in the study; even failing to distinguish between such basic statistical concepts as the median and the mean. The authors should include a statistician as a coauthor, or at least someone who has a basic understanding of the statistical methods used in the paper. Specific comments: p. 1 The authors state in the title that this is a retrospective study, in the abstract that it is a longitudinal study and in the consort statement that this is a cross-sectional study. p. 2 The results section of the abstract refers to P50 which usually represents the fiftieth percentile otherwise known as the median. The methods section refers to the "mean absence duration". Nowhere in the manuscript is the median mentioned. Perhaps it should be? What is actually presented, the mean or the median? p. 2 The first sentence in the conclusion of the abstract: "This longitudinal study used a unique and rich database to quantify SA duration by cause for certain MSK and MH-related SA events." This is not a conclusion of this study. The conclusion should relate to the aim of the study. p. 2 The second sentence in the conclusion of the abstract: "Our findings can be used by public health practitioners and healthcare managers to develop and implement tailored and targeted workplace interventions for employees with MSK and MH problems." I do not see how this conclusion can be reached from the results presented. I understand "targeted" in that one might want to focus on interventions to shorten longer SA spells, however this is not mentioned in the conclusion subsection of the paper. Regarding "tailored" see my response below on the conclusion subsection. p.3 Second point in the dotted list: "Survival analyses and Cox's proportional hazards models". This sentence makes it sound as if the Cox model does not belong within the discipline of survival analysis. p. 8 "The study population includes all participants in the EASY (Early Access to Support for You) SA management service of a Scottish health board".
---

	How many were these participants? Is it an inclusion criterion that is not mentioned explicitly that an individual need to have had at least one SA spell to be included? p.9. "Mean absence duration was calculated for the entire EASY population in the study" Does this include those that did not get any SA spell during the specified time period? p. 10 "Absence duration was analyzed using Kaplan Meier survival analyses and Cox's proportional hazards models to determine the hazard (risk) of absentees returning to work." This needs reformulation, which has been done in the abstract, but not here. p. 10 "11,694 individual people and 48,007 absences". How were the sickness absence spells/days distributed among the participants? Were there a few individuals that contributed most of the spells/days? p. 10 "We used a time-varying coefficient in our analysis to correct for non-compliance to the Cox assumption." What is meant by this sentence? What coefficient was time-varying? Non-compliance is specific term in clinical trials, which I do not think you are referring to here. There is no such thing as the "Cox assumption", perhaps you are referring to the proportional hazards assumption. It is not mentioned for which coefficient the proportional hazards assumption is not fulfilled. This needs to be explained much better. p. 12 "The three most common types of MSK problems in this population are low back pain (33.6%)..." 33.6% of what? Participants with MSK problems? SA spells due to MSK? SA days due to MSK? p. 16 "In terms of absence cause, in comparison to all of EASY (minus MSK & MH), time to RTW was 49% (95% CI .49-.52, P<0.0001) and 76% (95% CI .23-.66, P<0.0001) longer for absences due to MSK and MH conditions, respectively." This is the correct interpretation of the results of an accelerated failure time model. This is not a correct interpretation of the results from a Cox regression, where the HR is interpreted in terms of multiplying the hazards, NOT survival time. This erroneous interpretation is made throughout the section. p. 16 The point estimate of the HRs are presented as percent, but the confidence intervals as proportions. p. 16 Why use administrative services as the reference? This choice is not motivated. If using a reference at all, why not increase power by using the largest category Nursing/midwifery as reference? p. 21 "Our results suggest that employees with upper limb disorders and depressive symptoms in particular, require more tailored interventions to assist them in the return to work process following a SA event." You have determined that these sickness absence spells tend to be longer, but what in your results are saying that these SA spells benefit more from "tailored" interventions. This conclusion might be correct, but I cannot see how it is a result of this study. p.36 .25 and .75 are only partially visible in the graph.
--	---

VERSION 2 – AUTHOR RESPONSE

Reviewer: 1

Reviewer Name: Daniel Olsson

Institution and Country: Unit of Medical Statistics, Department of Learning, Informatics, Management and Ethics (LIME), Karolinska Institutet, Sweden Please state any competing interests: None

Please leave your comments for the authors below

Comment: The authors show that different health conditions have different length of sickness absence and that these differences are relatively large in magnitude. This is by no means controversial results and only requires simple descriptive statistics to show. However, the authors' conclusions go far beyond what can be substantiated by the results of the study.

The authors do not appear to understand the statistics presented in the study; even failing to distinguish between such basic statistical concepts as the median and the mean. The authors should include a statistician as a coauthor, or at least someone who has a basic understanding of the statistical methods used in the paper.

Author Response: We thank the reviewer for his comments. We have addressed all the specific points raised below and provided detailed answers on how we dealt with each point. Concerning the reviewer's comment that our results are not 'controversial', we agree with the reviewer and did not make any such claim. However, we would also argue that for research to be important and have impact- both for this occupational group but also wider impact extending to society- it does not need to be controversial, but rather to be research of high standard adding to knowledge by addressing an evidence gap. Sickness absence (SA) is a significant public health burden on government, employees, employers, and public resources. This is due to loss of productivity, increased workload on other staff, as well as resources spent to cover incapacity to work and restore health. The public health burden of sickness absence is recognized and governments and employers are developing policies toward providing support to employers and employees in improving health, reducing sickness absence rates, and improving return to work (RTW) times. The healthcare sector exhibits one of the highest SA rates. Despite a great deal of research recognizing the prevalence of MSK and MH issues and their impact on SA rates, there remains a lack of information - often due to the lack of detailed and reliable data - on how certain types of MSK and MH health conditions affect SA duration. This knowledge is vital to inform and improve current and new SA interventions, as well as support workplace modifications for SA prevention. Our study fills this knowledge gap, by using a unique and large database, giving six full years of sickness absence and return to work data. The range of variables collected - including demographic, job, SA start and end dates, self-reported conditions, allowed for the investigation of SA duration and time to return-to-work for healthcare employees by age, gender, job and health conditions. Furthermore, simple descriptive statistics would not be able to assess the variables significantly impacting RTW times.

We have responded to the reviewer's points on the statistical nomenclature in the specific points below (e.g. regarding mean, median). We have also sought further advice from our colleague Dr Daniel F Mackay (Reader in Public Health). Daniel Mackay is a Reader in Public Health and has extensive experience of statistical and econometric methods, especially time series methods, the evaluation of complex interventions and epidemiology. Dr Mackay has now joined the author team and we have both a senior (Mackay) and junior (Bhaskar) statistician who have reviewed the methodology, analysis and paper. Please note following Dr Mackay's review, he suggested we make a small amendment to our STATA code which very slightly changed some of the values of estimates in the second decimal place but the results did not change (the smallest change was 0.01 and the biggest change was 0.133 to one of the p-values).

Specific comments:

p. 1 The authors state in the title that this is a retrospective study, in the abstract that it is a longitudinal study and in the consort statement that this is a cross-sectional study.

Author Response: The cross-sectional wording in the consort statement was an error. We thank the Reviewer for spotting this error and we have corrected it. We believe that our title "Evaluating Sickness Absence Duration by Musculoskeletal and Mental Health Issues. A retrospective cohort study of Scottish Healthcare Workers." best describes our study and have ensured that this description is consistent throughout.

p. 2 The results section of the abstract refers to P50 which usually represents the fiftieth percentile otherwise known as the median. The methods section refers to the "mean absence duration". Nowhere in the manuscript is the median mentioned. Perhaps it should be? What is actually presented, the mean or the median?

Author Response: The result section of the abstract does not only refer to the P50/median results of the length of absence and return-to-work times, but to the average/mean times as well and we have made the distinction very clear when we are referring to the average and when to the P50. So as not to cause any confusion, we have further described the P50 as the median both in the abstract and the main document (please see abstract and page 13; last paragraph). Please also note that we have corrected a typo in the average MSK absence duration that we spotted.

p. 2 The first sentence in the conclusion of the abstract: "This longitudinal study used a unique and rich database to quantify SA duration by cause for certain MSK and MH-related SA events." This is not a conclusion of this study. The conclusion should relate to the aim of the study.

Author Response: We have amended our conclusion section (please see abstract; conclusion section).

p. 2 The second sentence in the conclusion of the abstract: "Our findings can be used by public health practitioners and healthcare managers to develop and implement tailored and targeted workplace interventions for employees with MSK and MH problems." I do not see how this conclusion can be reached from the results presented. I understand "targeted" in that one might want to focus on interventions to shorten longer SA spells, however this is not mentioned in the conclusion subsection of the paper. Regarding "tailored" see my response below on the conclusion subsection.

Author Response: We have amended our conclusion section (please see abstract; conclusion section).

p.3 Second point in the dotted list: "Survival analyses and Cox's proportional hazards models". This sentence makes it sound as if the Cox model does not belong within the discipline of survival analysis.

Author Response: We have amended the wording here and the second point now reads: "Cox's proportional hazards models applying time varying coefficients were used to estimate SA duration due to musculoskeletal (MSK) and mental health (MH) problems over six years (N=48,007 unique SA events)."

p. 8 “The study population includes all participants in the EASY (Early Access to Support for You) SA management service of a Scottish health board”. How many were these participants? Is it an inclusion criterion that is not mentioned explicitly that an individual need to have had at least one SA spell to be included?

Author Response: We have specified in the study population section that the study population includes all participants in the EASY services over 6 years and that the data includes all those employees who voluntarily engage with the EASY service. A very brief description of EASY is given, that specifies this is a SA management service starting on the first day of an absence event. Furthermore, we have referred the readers to our previous publications on the EASY service for more detailed information. The total number of individuals (11,694 unique employees) as well as the total number of absence events (48,007) in the May 2008-April 2014, are reported in the first sentence of our Results section (please see page 10; Results section).

p9. “Mean absence duration was calculated for the entire EASY population in the study” Does this include those that did not get any SA spell during the specified time period?

Author Response: Please also see the response to the above point. EASY is a sickness-absence management service that employees can access voluntarily and starts on their first day of absence. As this is a study examining sickness absence duration, any employees without a SA event would not be in the EASY database and with no SA event could not be included in the analysis.

p. 10 “Absence duration was analyzed using Kaplan Meier survival analyses and Cox’s proportional hazards models to determine the hazard (risk) of absentees returning to work.” This needs reformulation, which has been done in the abstract, but not here.

Author Response: Amended.

p. 10 “11,694 individual people and 48,007 absences”. How were the sickness absence spells/days distributed among the participants? Were there a few individuals that contributed most of the spells/days?

Author Response: As this is a study over 6 years, it is expected that most individuals will have multiple absence events. Most SA events are of short duration with 73% of all absence events lasting ≤ 10 days and 87% ≤ 1 month (30 days). Furthermore, 77% of all individuals had 1-5 absences (averaging less than one absence per year); and only 4% of individuals had more than 10 absences over the six year period. In our survival and Cox regression analyses, we have accounted for multiple absences per individual.

p. 10 “We used a time-varying coefficient in our analysis to correct for non-compliance to the Cox assumption.” What is meant by this sentence? What coefficient was time-varying? Non-compliance is a specific term in clinical trials, which I do not think you are referring to here. There is no such thing as the “Cox assumption”, perhaps you are referring to the proportional hazards assumption. It is not mentioned for which coefficient the proportional hazards assumption is not fulfilled. This needs to be explained much better.

Author Response: Yes, the reviewer is correct that the proportional hazards assumption was violated and to correct for this we had included time-varying coefficients for the variables for which the assumption was not met. To clarify this and also for increased transparency for the reader we have modified our methods section to describe the test used to check the proportional hazards assumption section but also included in the results the specific variables for which the assumption was not met.

Table 2 now includes the time-varying coefficients as well and Table S1 in the Supplementary Material includes the results of the proportional hazards test for all models and all the variables.

p. 12 “The three most common types of MSK problems in this population are low back pain (33.6%)...” 33.6% of what? Participants with MSK problems? SA spells due to MSK? SA days due to MSK?

Author Response: This refers to the MSK sickness absence events due to low back pain as a proportion of all SA events. We have clarified this in the main manuscript (please see page 12; 2nd paragraph).

p. 16 “In terms of absence cause, in comparison to all of EASY (minus MSK & MH), time to RTW was 49% (95% CI .49-.52, $P < 0.0001$) and 76% (95% CI .23-.66, $P < 0.0001$) longer for absences due to MSK and MH conditions, respectively.” This is the correct interpretation of the results of an accelerated failure time model. This is not a correct interpretation of the results from a Cox regression, where the HR is interpreted in terms of multiplying the hazards, NOT survival time. This erroneous interpretation is made throughout the section.

Author Response: This wording has been corrected and we have included the hazards ratios throughout this description to ensure clarity for the readers.

p. 16 The point estimate of the HRs are presented as percent, but the confidence intervals as proportions.

Author Response: The hazard ratio (HR) values have been added (please also see response to the above comment).

p. 16 Why use administrative services as the reference? This choice is not motivated. If using a reference at all, why not increase power by using the largest category Nursing/midwifery as reference?

Author Response: We appreciate the reviewer’s comment that most often, the largest category is used as the reference category and we have re-run the model using the Nursing/Midwifery (i.e. the largest category) as the reference category. Please note that the results do not change.

p. 21 “Our results suggest that employees with upper limb disorders and depressive symptoms in particular, require more tailored interventions to assist them in the return to work process following a SA event.” You have determined that these sickness absence spells tend to be longer, but what in your results are saying that these SA spells benefit more from “tailored” interventions. This conclusion might be correct, but I cannot see how it is a result of this study.

Author Response: We have amended the wording to address the Reviewer’s comment (please see page 2; 2nd paragraph).

p.36 .25 and .75 are only partially visible in the graph.

Author Response: We gather that the Reviewer is referring to the 95% CIs in the graph. The CIs are very small. We have added a separate Figure 2 with coloured CI lines that can be used for the online version of the paper.